# Assessments of Bacterial Community Shifts in Sediments along the Headwaters of São Francisco River, Brazil

**Marcos de Paula, Jr.** 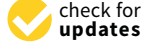**, Thiago Augusto Costa Silva** , **Amanda Soriano Araújo** and **Gustavo Augusto Lacorte** *

Molecular Biology Lab, Department of Science and Languages, Federal Institute of Minas Gerais–Bambuí Campus, Bambuí 38900-000, Brazil; prof.marcosbiologo@gmail.com (M.d.P.J.); thacs2006@yahoo.com.br (T.A.C.S.); amanda.barezani@ifmg.edu.br (A.S.A.)

* Correspondence: gustavo.lacorte@ifmg.edu.br; Tel.: +55-37-3431-5600

**Abstract:** Sustainable use of freshwater resources for human civilization needs requires the assessment and monitoring of freshwater health, and bacterial communities from riverbed sediments have been shown to be susceptible to chronic anthropogenic disturbances in freshwater ecosystems. Here, we took advantage of the occurrence of well-recognized adjacent sections from the Upper São Francisco River basin with well-recognized levels of anthropogenic activity intensity to test the applicability of sediment bacterial communities as bioindicators of impacts on freshwater ecosystems. We applied 16S amplicon sequencing to estimate the diversity and composition of bacterial communities from 12 sampling sites across the Upper São Francisco River basin, classified as being of no, low, or high intensity of anthropogenic activities, and used diversity metrics and LEfSe to compare the patterns of community structure. Our results revealed that accessed sediment environments associated with land areas with a high intensity of anthropogenic activities presented the lowest levels of community diversity, and the bacterial community compositions of these environments were significantly different from the other sampled areas. Our findings can be considered a source of evidence for the usefulness of bacterial community-based approaches as a tool for diagnosis and monitoring of ecosystem health in areas of vulnerable freshwater environments, and can even be incorporated into regular water quality programs.

**Keywords:** freshwater; river; microbes; community; ecosystem

## 1. Introduction

Freshwater ecosystems (in rivers, wetlands, and lakes) support huge biodiversity and provide environmental goods and services of critical importance to human populations everywhere [1]. In the last few decades, freshwater ecosystems have been severely impaired by human activities, such as overexploitation (large irrigation projects), alterations of natural watercourses (dam construction), degradation of water quality by pollution and eutrophication (industrial, agriculture, and urban discharges), while human societies' dependence on freshwater goods and services is still increasing [2,3]. This challenge of supporting human needs with a livable impact on freshwater ecosystems requires the adoption of sustainable water management strategies, and the assessment and monitoring of freshwater health approaches constitutes one of the basic management steps [4]. Assessment and monitoring approaches are based on analyses of ecological indicators (physical, chemical, and biological) developed to detect and measure changes in freshwater ecological conditions [5,6].

Biological indicators have been increasingly used in management programs because they are highly sensitive to environmental stressors [7], and the classic taxonomic groups used as bioindicators include mainly aquatic macroinvertebrates, fishes, waterbirds, and plants because microorganisms present a considerable limitation in taxonomic identification by classical methods [8]. However, with the consolidation of culture-independent methods for bacterial identification in recent years, it has been possible to identify the bacterial

groups rapidly and accurately [8–11]. These molecular approaches have made the use of bacterial communities as bioindicators an attractive and viable alternative because they have several advantages as freshwater ecological indicators: they comprise the majority of aquatic biomass (ubiquitous, abundant, and diverse); they play key roles in aquatic ecosystem functioning (nutrient cycling, photosynthetic energy source); and they are highly sensitive to physical, chemical, and biotic environmental disturbances [9,12–14]. Briefly, the use of bacterial communities as bioindicators for aquatic ecosystems is based on the premise that bacterial diversity is a key factor affecting the biological quality of ecosystems due to its role in nutrient recycling, the degradation of pollutants, and the stability of ecosystems. Thus, changes in taxonomic diversity and the composition of bacterial communities is a suitable indicator of perturbations within an ecosystem [14–19]. In this context, massive sequencing of DNA amplicons considered to be taxonomic markers (ribosomal genes) derived from environmental matrices can be used to determine the taxonomic diversity and composition of bacterial communities [20].

Considering the matrices available in river environments to sampling, recent studies have shown that bacterial communities from water and sediment are distinct components of aquatic ecosystems (a stream tends to create a more dynamic environment for water), of which the bacterial diversity in sediments is considerably higher and more suitable to predict chronic environmental disturbances caused by human activities than water bacterial communities [21–26]. Recent evidence has revealed that during the intensification of land-use processes, contaminants are absorbed by the fine particles of sediments and interact with local bacterial communities, causing long-term impacts on the biological organization of the communities and leading to changes in both the diversity and composition of the bacterial communities [27–30].

São Francisco River is the fourth largest river in Brazil, with a total length of 2830 km and a catchment area of 639,219 km$^2$. The river basin crosses the two most populous regions of Brazil, whose activities impact the river basin ecosystems both by the direct use of the water (urban water supply, irrigation, hydroelectric power generation) and by the intensive land-use of the areas closely associated with the basin by urbanization, intensive agriculture and livestock production, mining, and industrial plants [31]. Given the importance of the river basin for local populations and the recognized impacts caused by direct and indirect human activities on the São Francisco river basin, a government plan for monitoring the water quality of the São Francisco river basin was implemented and, since 1997, water quality has been monitored by classic physicochemical and biological parameters, which are measured quarterly [32–34].

For this study, we took advantage of the occurrence of contiguous São Francisco river areas well-recognized (by historical monitoring data) as non-impacted, with low and high intensities of land-use by human activities, to test the applicability of sediment bacterial communities as health bioindicators of the São Francisco river ecosystems. Specifically, our main goals were (i) to verify if changes in the diversity and structure of sediment bacterial communities are associated with the intensification of land-use and their impacts, and (ii) to identify the major sensitive bacterial taxa (i.e., indicator taxa), which could be used to detect environmental changes in the São Francisco river basin.

## 2. Materials and Methods

### 2.1. Study Sites and Sampling

Besides the socio-economic importance of the São Francisco river basin, we selected the first portion of the São Francisco river basin (named the SF1 microbasin) as a study model due to two practical criteria. First, the water quality parameters of the SF1 microbasin have been monitored by the Minas Gerais State Water Resources Institute (IGAM, 2020) and the historical record of this monitoring program (available since 1997) provided us with reliable parameters to identify the SF1 sections with distinct intensity levels of anthropogenic land-use. Secondly, although there are other microbasins of the São Francisco river basin with available monitoring data, the SF1 microbasin is the only one that has contiguous

areas with increasing levels of land-use intensity, and that starts from an area with no anthropogenic impact that corresponds to the spring area located at a conservation unit with permanent preservation (Figure 1). Furthermore, we selected collection points only at the São Francisco river course, since the main river course receives inputs from the other tributaries of the basin.

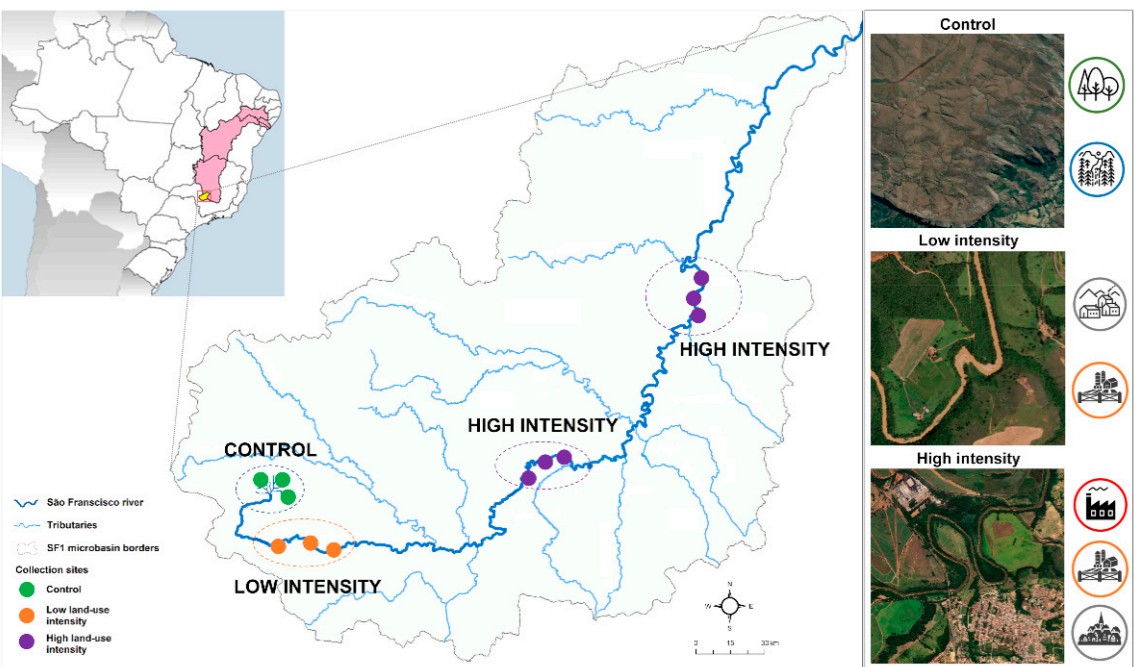

**Figure 1.** Sampling areas and collection sites in the Upper São Francisco SF1 microbasin.

So, the sampling area selected as the control is located within the Serra da Canastra National Park, a conservation unit with permanent preservation, by which it has the key role of protecting the São Francisco River springs. In order to systematically identify the sampling areas with low and high land-use intensity, we accessed the monitoring program database and selected the water quality parameters most impacted by the main anthropogenic activities registered for SF1 (domestic and industrial sewage as well the leaching of agricultural and livestock inputs). Then, we tested if there were significant differences for these parameters among the first three monitoring areas. Based on this analysis, we were able to identify that the first area presented evidence of a low intensity of land-use impacts, and the two subsequent areas presented evidence of a high intensity of land-use activities (Figure 2). There were 3 collection sites established for each sampling area, totaling 12 sediment collection sites. For each collection site, 50 g of surface riverbed sediment was collected with a sterile bottle, at a distance of approximately 5 m from the river border or at the middle point of the mainstream. After sampling, the bottles were stored in a portable ice box and transferred to a laboratory within 12 h. All samples were collected in 2017 during the dry season. The access to genetic material related to this study was properly registered in the official database of the Brazilian genetic patrimony (locally named SISGEN), with access number A07CD46.

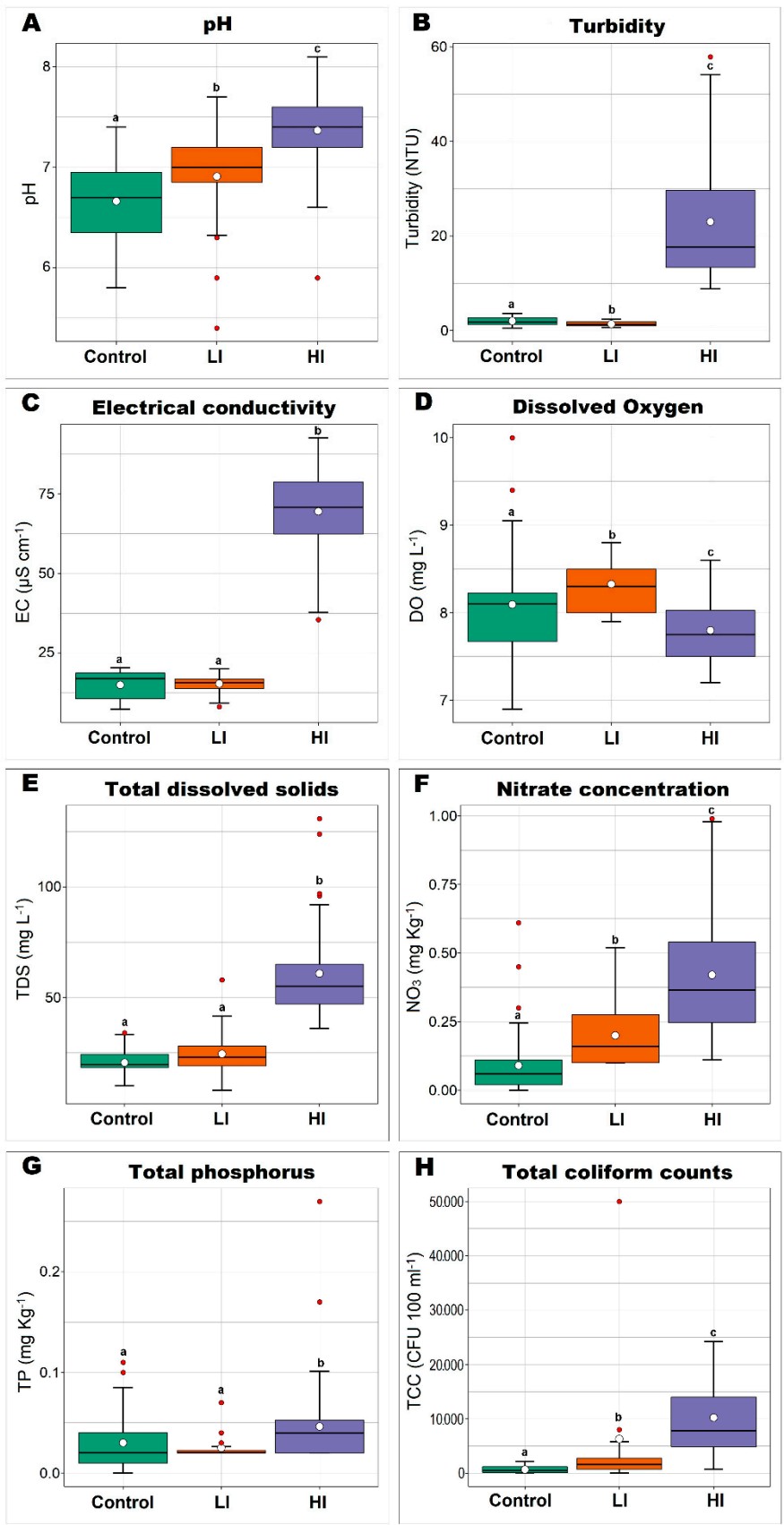

**Figure 2.** Boxplot of water quality parameters in the mainstream measured quarterly over the last 5 years until the collection. Different lowercase letters indicate significant differences ($p < 0.05$) in mean values (Kruskal-Wallis test, performed in R software, version 3.3.2).

## 2.2. DNA Extraction, PCR Amplification, and High-Throughput Sequencing

The total genomic DNA was extracted from the sediment samples using a DNeasy PowerSoil Kit (Qiagen, Hilden, Germany) according to the manufacturer's protocol. The samples were processed right after they came from the collection step. DNA concentrations were estimated using a NanoDrop Spectrophotometer (Thermo Scientific, San Jose, CA, USA), and stored at −20 °C for further analysis.

The bacterial diversity of each sediment sample was accessed by high-throughput sequencing of the amplified V3–V4 region of the 16S rRNA gene by using primers 314F (CCTAYGGGRBGCASCAG) and 806R (GGACTACNNGGGTATCTAAT) [20], following the guidelines established by the Earth Microbiome Project [35]. PCR amplifications were performed in triplicate using customized primers containing both Illumina adapters and distinct barcode sequences so that each amplified DNA sample contained a different combination of barcodes in order to distinguish the libraries after the sequencing step [35]. All PCR reactions were carried out using the optimized PCR reaction mix OneTaq® Hot Start Quick-Load® 2X Master Mix with GC Buffer (New England Biolabs), with the following thermocycling PCR program: 94 °C for 3 min; 25 cycles of 94 °C for 45 s, 50 °C for 60 s, 72 °C for 90 s, and 72 °C for 10 min. The final PCR reactions were cleaned up using AMPure XP beads (Beckman Coulter, Brea, CA, USA) and quantified with Picogreen dsDNA assays (Invitrogen, Eugene, OR, USA). In an attempt to normalize the sequencing step, equal amounts of PCR products from each sample (50 ng/sample) were pooled. Pooled PCRs were subjected to electrophoresis on 1% agarose gel for purification by isolation of the PCR bands (300–500 bp) using a sterile razor. Pooled PCR bands were purified from agarose gel with NucleoSpin™ Gel and a PCR Clean-up Kit (Macherey-Nagel™, Düren, Germany) and quantified by a Qubit®2.0 Fluorometer (Thermo Scientific, CA, USA). Pooled PCRs were subjected to paired-end sequencing (2 × 250 bp) on a MiSeq platform (Illumina, San Diego, CA, USA) with a MiSeq Reagent Kit V2 (500 cycles) from the CEFAP Facility (São Paulo University, São Paulo, Brazil). All sequencing data generated in this study can be accessed from the GenBank Database at Bioproject PRJNA611749.

## 2.3. Data Analyses

The amplified 16S rRNA gene sequences were processed using the analytical bioinformatic pipeline QIIME 2 (Quantitative Insights Into Microbial Ecology, version 2), version 2019.10 pipeline [20,36]. The. plugin "demux" was used to visualize interactive quality plots and check the read quality. The plugin "DADA2" (abbreviation of Divisive Amplicon Denoising Algorithm, version 2) [37] was subsequently applied to remove primers, truncate poor-quality bases based on the interactive plots, dereplicate, identify chimeras, and merge paired-end reads. The representative sequences of the amplicon sequence variants (ASVs) were taxonomically assigned with a Naïve Bayes Classifier trained with the "feature-classifier" plugin using the 16S rRNA gene database, at a 99% similarity to the SILVA database (v.132), as a reference [38]. Exploratory and statistical data analyses were performed at the ASV level since the ASV approach is a higher-resolution equivalent of the operational taxonomic unit (OTUs), delineated by 100% sequence similarity [39,40].

In order to estimate if the changes in diversity and composition of the sediment bacterial communities are related to increasing land-use intensity by human activities, the samples were grouped by category of land-use intensity: control, low, and high-intensity land-use, coded here as C, LI, and HI areas, respectively. We used QIIME2 to generate alpha and beta-diversity vectors after rarefaction of the samples to 5000 sequences based on rarefaction curves generated previously (see Supplementary Material, Figure S1). For alpha-diversity analyses, Shannon diversity, Faith phylogenetic diversity, and Pielou evenness indices were calculated for each group, compared using Kruskal–Wallis pairwise tests, and visualized using boxplot graphs. Compositional patterns (beta-diversity) were estimated by Bray–Curtis and Unifrac (Weighted and Unweighted) dissimilarity metrics, tested with Permutational Multivariate Analysis of Variance (PERMANOVA), and visualized using

principal coordinate analysis plots (PCoAs). All alpha and beta-diversity comparative analyses were performed in Qiime2.

We used Qiime2 to generate a heatmap, including the most abundant families found in all samples, which compose >3% of the reads in at least one sample. We also performed linear discriminant analysis effect size (LEfSe) analysis to find bacterial indicator taxa specific to studied groups, using a Least Discriminant Score (LDA Score) of >4 as a cut-off parameter [41]. The LEfSe analyses were performed in Galaxy web applications available at https://huttenhower.sph.harvard.edu/galaxy/ (accessed on 1 October 2020).

## 3. Results

### 3.1. Diversity Indices of Sediment Bacterial Communities from Different Impacted Areas

The high-throughput sequencing generated a total of 211,884 sequence reads (ranging from 13,233 to 21,557), with the DADA2 approach recovering 1919 ASVs. The rarefaction curves indicated the saturation stage for all samples, suggesting that the sequencing depth was sufficient to cover most of the taxonomic diversity for all the accessed bacterial communities (Figure S1). The alpha-diversity indices calculated for each of the three groups of sediment samples evaluated presented the same variation pattern: all alpha-diversity indices calculated for areas of high-intensity land-use (HI) were significantly lower when compared to both the control (C) and low-intensity land-use (LI) areas (Figure 3). We also observed that there was an increase in the indices when comparing the C area (which corresponds to the river springs area) with the LI area, but without a consensus between the metrics used when tested by Kruskal–Wallis tests. We did not observe significant differences for all alpha-diversity indices between the samples of the two high-intensity land-use areas.

The plots of the principal coordinate analysis (PCoA) presented similar patterns of bacterial community composition (beta-diversity) for all applied metrics (Figure 4), with the two principal axes explaining between 43.4% (Bray–Curtis) and 68.5% (weighted Unifrac) of the total variation in the bacterial community structure. Bacterial communities were clustered in the ordination plot according to sediment group (C, LI, or HI). However, the PERMANOVA tests showed that sediment bacterial communities from the HI areas presented a taxonomic composition significantly different from the other sampled areas, but the sediment samples from the C and LI areas presented a similar taxonomic composition. Similar to the alpha-diversity analyses, we could not verify significant differences in taxonomic composition among sediment samples from the two areas of high-intensity land-use.

### 3.2. Distribution Pattern of Bacterial Taxa in Studied Areas

Based on the trained classifier implemented in QIIME2, the dominant phyla (with a relative abundance of >1% in all samples) were Proteobacteria (mainly Gammaproteobacteria), Firmicutes, Verrucomicrobia, Acidobacteria, Cyanobacteria, Actinobacteria, Bacterioidetes, Planctomycetes, and Cloriflexi (Figure 5). Proteobacteria, Firmicutes, Plactomycetes, and Bacterioidetes phyla were ubiquitous, but the distribution of the other dominant bacterial phyla varied among the sediment samples.

At the family level, we found 20 bacterial families with a relative abundance of >3% in at least one sediment sample. Remarkably, the distribution patterns for some of these dominant families were associated (positively or negatively) with the land-use intensity groups (Figure 6). The families of Pedosphaeraceae and Pseudomonadaceae, and an uncultured group of Acidobacteriales, were positively associated with sediment samples from the control area. Alternatively, the abundance of the Micrococcaceae, Chitinophagaceae, and Sphingomonadaceae families was negatively related to the control samples. On the other hand, the families of Anaerolinaceae and Synthophaceae were positively associated with sediment samples from high-intensity land-use areas, whereas Beijerinckiaceae was negatively related. We also found ubiquitous families, such as Burkholderiaceae, Bacillaceae, Archangiaceae, Xanthobacteriaceae, and Clostridiaceae.

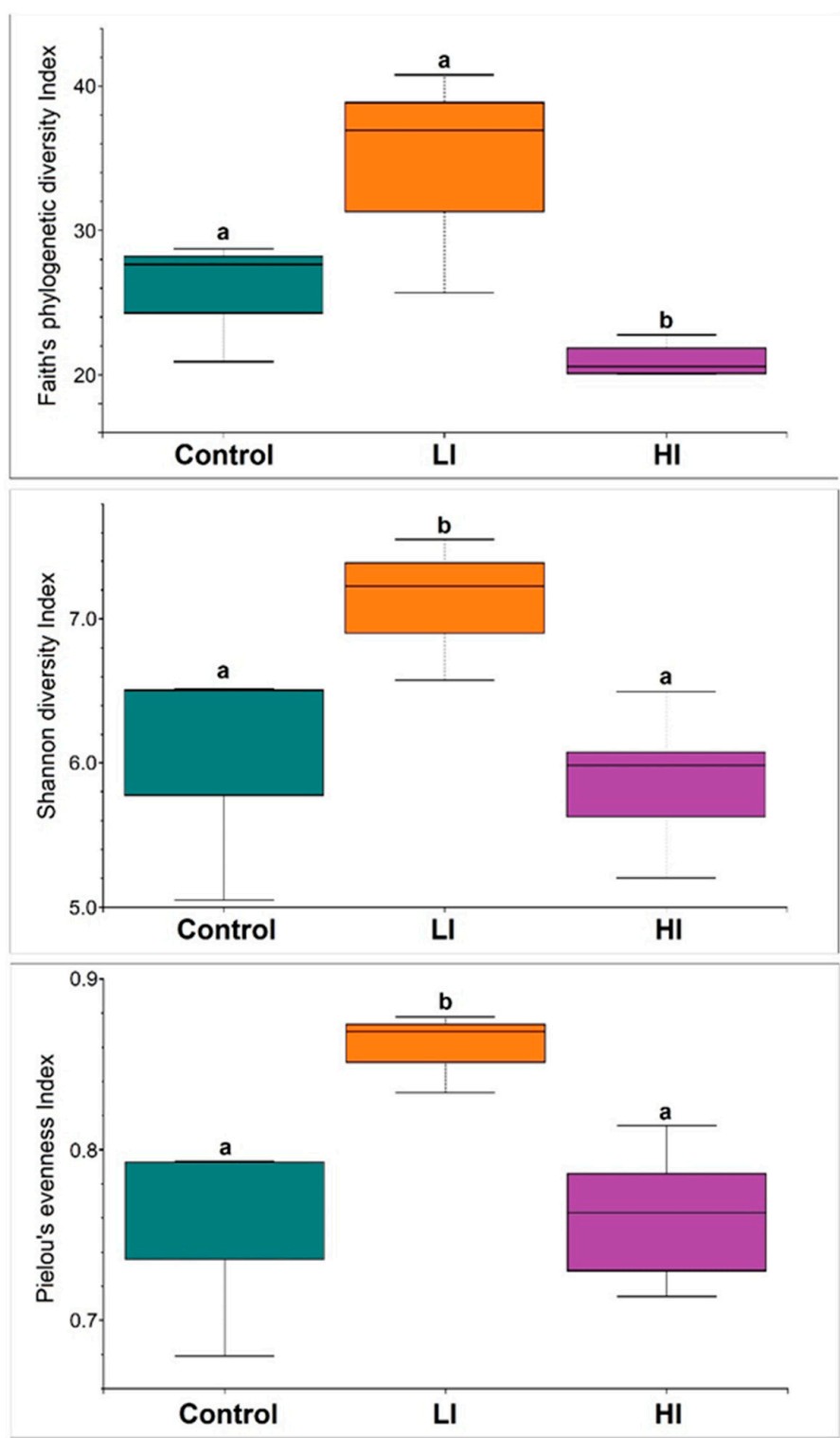

**Figure 3.** Comparisons of alpha-diversity indices (Faith, Shannon, and Pielou) among sediment bacterial communities from three sampled areas: control, low, and high-intensity land-use. Different lowercase letters indicate significant differences ($p < 0.05$) in mean values (Kruskal–Wallis test).

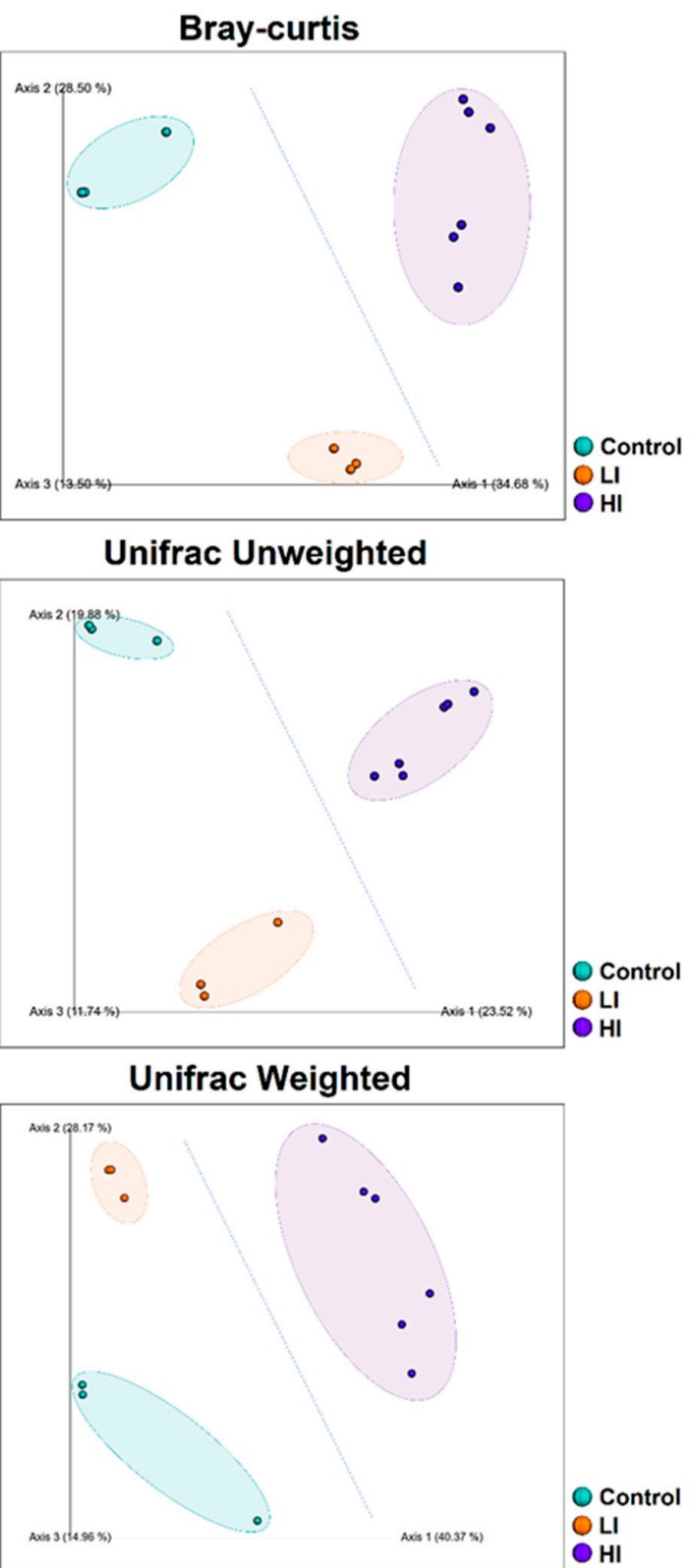

**Figure 4.** PCoA plots for each dissimilarity metric applied (Bray–Curtis, unweighted Unifrac, and weighted Unifrac) showing the clustering patterns of sediment community samples. The blue dotted line represents the dissimilarity pattern (control + LI vs. HI samples) supported statistically by the PERMANOVA tests.

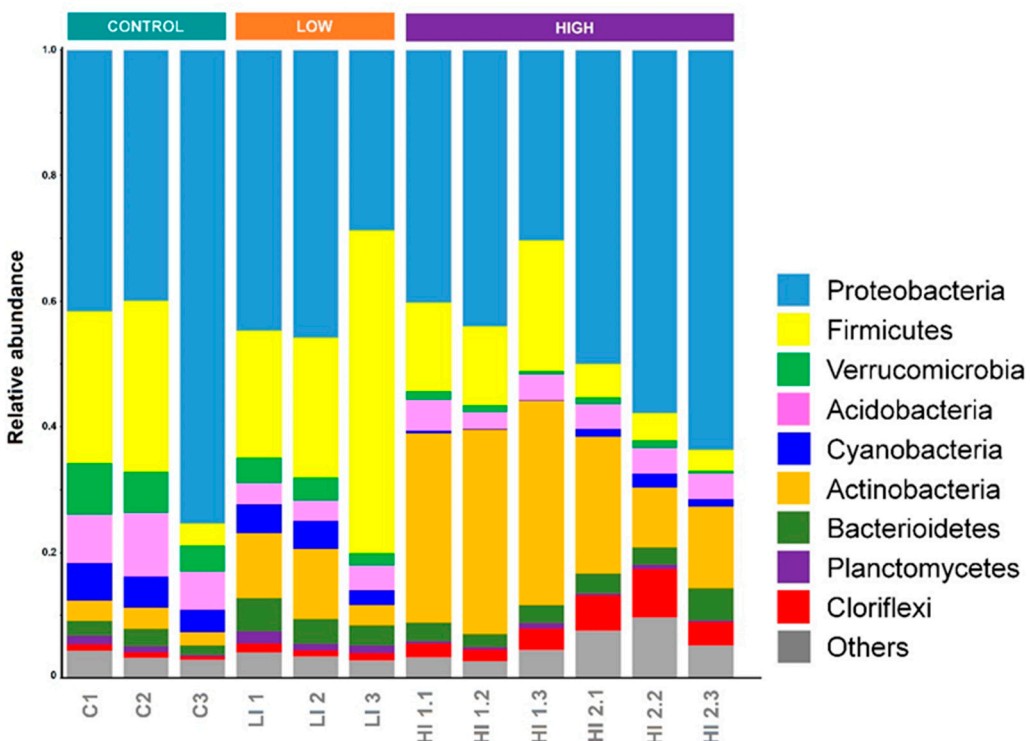

**Figure 5.** Relative abundance of sediment bacterial communities at a phylum level for each sample.

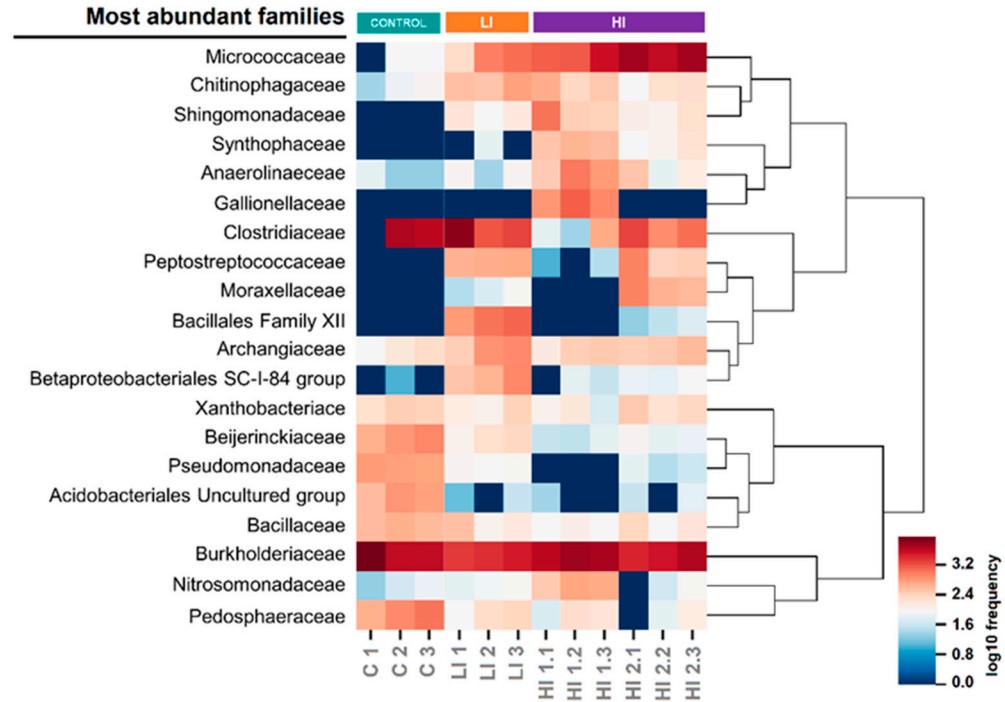

**Figure 6.** Heatmap plot showing the relative frequency of the most abundant bacterial families in each sediment sample.

In this study, we performed LEfSe analysis in order to search dominant taxa with LDA scores of ≥4, which recovered 45 bacterial taxa associated with one of the three groups of sediment samples evaluated (20, 13, and 12 for C, LI, and HI, respectively) (Figure 7). Lineages of the Massilia genus were strongly associated with the control sediment group (with an LDA score of 5). Similarly, lineages of Exiguobacterium were associated with the

LI group. Five lineages were closely related to the HI sediment group, three assigned to the Micrococcaceae Family and two assigned to the Actinobacteria phyla.

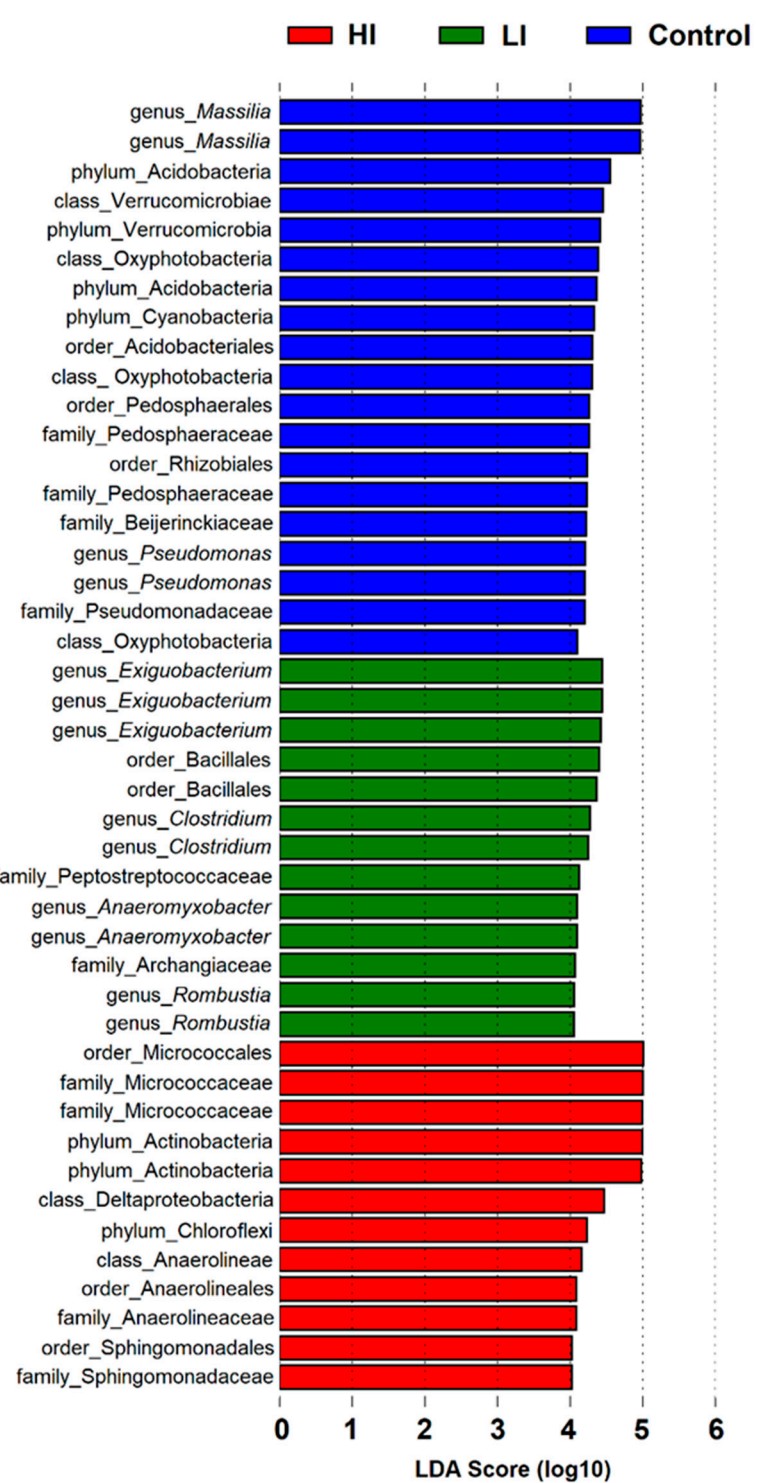

**Figure 7.** LDA values for indicator bacterial taxa in the three groups of sediment samples generated by LEfSe analysis.

## 4. Discussion

One of our main goals was to verify if the variation on patterns of taxonomic diversity (alpha and beta-diversity) are associated with changes in land-use intensities, and our estimates of alpha-diversity indices revealed two relevant patterns. First, there was an

increase in diversity when we compared the sediment communities of the control area with sediment communities from low-intensity land-use areas. This pattern could be associated with the riverbed composition of the control sampling area (river springs), which consists of the rocks from which the spring emerges, specifically layers of fine sediment particles and organic matter much less thick than is found in the riverbed samples from the low-intensity land-use areas. Second, we found that for high-intensity land-use areas, the diversity indices decreased significantly to levels even below those found for the spring areas, even with riverbed compositions similar to low-intensity land-use areas. This pattern is in accordance with the theoretical premises and observational evidence that established that as microorganisms in a community establish various ecological relationships with biotic and abiotic local conditions, the intense use of land by human activities, such as the disposal of chemical components as well as the inoculation of exogenous microorganisms, significantly alter the selective pressures on the local microbiota, culminating in a loss of diversity [12–14,23,42,43]. Furthermore, maintaining biodiversity in freshwater ecosystems is crucial to increase ecosystem resilience in the face of the current scenario of persistent environmental disturbances, since greater biodiversity reflects more ecological equivalents in the ecosystem to assume the key ecological functions which maintain the ecosystem health [22]. Thus, the use of an approach that estimates the bacterial diversity of the key communities of a freshwater ecosystem is also a viable alternative to estimate the fragility and resilience of an impacted environment.

In addition to presenting lower levels of diversity, we also found that the bacterial community composition of sediment samples from high-intensity land-use areas was significantly different from the other sampled areas. A similar pattern was found by other studies in which different types of human impacts caused changes in the composition of the sediment communities [17,18,44,45]. Our results reinforce the potential of the applicability of sediment community composition as a parameter of land-use intensity bioindication. Moreover, as high-throughput sequencing procedures become more accessible and cheaper, the use of these sediment bacterial communities as bioindicators becomes more attractive.

Our second main goal was to search for bacterial indicator taxa in sediment communities, which could be used to detect environmental changes. At a phylum level, we found that Proteobacteria was the most abundant phyla for all the sediment samples and the other nine dominant phyla. This pattern of bacterial phyla dominance has been frequently reported in studies of bacterial diversity in riverbed sediment and other wetland soils, and so, we consider phylum-level analyses to have low potential as health bioindicators for the ecosystems studied here [19,23,24,46–48]. Our results show that there was a clear similarity of taxonomic composition between the control and LI bacterial communities. It was also verified that there was a significant turnover of bacterial sediment communities from LI to HI sampled areas. Typical inputs derived from farming practices (fertilizers, pesticides, and livestock residues) that occur at NP sampled areas could be creating selective local pressures capable of becoming a bacterial composition of HI sediment communities different from the other sampled areas. Several studies have shown that higher concentrations of organic and inorganic nutrients associated with agricultural activities may alter microbial communities and their functions on river streams and sediment [49–53]. Considering the dominant bacterial phyla, we found that some phyla presented abundance associated with occurrence area, especially if the area presented land-use restrictions, in which Actinobacteria and Chloriflexi phyla occurred in greater abundance at HI areas, while Verrucomicrobia and Cyanobacteria phyla were less frequent at NP areas. Several studies also found similar results; some of them presented a correlation of these phyla with an increase in nitrogenous and phosphorous nutrients, whose origin was related to agricultural or urban input runoff [29,54–56].

At the family level, we found some dominant bacterial families with distribution patterns that contributed significantly to the dissimilarity among the sediment communities, associating (positively or negatively) with the land-use intensity groups, and thus, presenting considerable potential as bioindicators. The Anaerolinaceae and Synthophaceae

families were positively associated with sediment samples from high-intensity land-use areas. In contrast, Pedosphaeraceae and Pseudomonadaceae were positively associated with samples from spring areas, where there is no evidence of human land-use. Xie et al. [29] also correlated lineages of Anaerolinaceae and Synthophaceae families to sediment samples from high human-impacted areas as well Pedosphaeraceae lineages to samples of low impact of China rivers, reinforcing the potential of these three families as biomarkers of environmental quality. On the other hand, the families of Burkholderiaceae, Bacillaceae, Archangiaceae, Xanthobacteriaceae, and Clostridiaceae were dominant in all sediment samples and have low worth as bioindicators due to the ubiquity presented by these families. Moreover, the Micrococcaceae, Chitinophagaceae, and Sphingomonadaceae families were negatively correlated with sediment samples from spring waters. Sphingomonadaceae are often found in high proportions in habitats contaminated with recalcitrant (poly) aromatic compounds, and it has been considered to be an indicator of the degree of urbanization [57,58]. Roberto et al. [25] reported a high abundance of Chitinophagaceae in sediment samples from a highly impacted urban stream. Based on this evidence, it is possible that the urbanization process could be the origin of these organisms in sediment communities.

The LEfSe results show eight bacterial taxa with a high potential bioindicator of high land-use intensity. Three of these ASVs were identified as belonging to the Micrococcaceae family and two were assigned to the Actinobacteria phyla. In addition, two ASVs belonging to the genus Massilia (Burkholderiaceae family) show a strong association with samples from the control environments. The ASVs with the highest potential for prediction of low land-use intensity correspond to three bacterial lineages of the Exiguobacterium genus. However, the use of specific bacterial taxa as ecological indicators is an approach that needs to be considered with more caution than the methods based on the patterns of diversity of the whole bacterial community [8,10]. Here, we consider that the identification of all ASVs with a high correlation with the three categories of environments, revealed by LEfSe results, constitutes the first step of a validation process. Additional evidence of metabolic characteristics of these taxa, their ecological roles in each ecosystem, as well the verification of the same patterns of variation in similar systems are required for the complete validation of these bacterial taxa as health indicators of freshwater environments.

In this study, we characterized variation patterns of sediment bacterial communities from the SF1 microbasin. In addition to this region constituting a great model system for anthropogenic impacts on freshwater ecosystems, the SF1 microbasin also has socio-economic and ecological importance for the entire São Francisco river basin, and the health of these environments must be constantly monitored and managed. The current monitoring method of freshwater quality in Brazil was defined by CONAMA Resolution 357/2005 [33], which is based, with local adaptations, on the Water Quality Index (WQI), originally developed by the National Sanitation Foundation in the United States to estimate the quality of water for human consumption purposes [59]. Due to the fact that the parameters used to estimate the WQI index were not originally designed to assess the health of freshwater ecosystems, it is reasonable to consider that new parameters, which estimate the health of ecosystems more efficiently, should be incorporated into monitoring programs of the São Francisco river basin, including the recently developed microbiological bioindicators. Moreover, some studies have reported that bacterial communities sampled from water bodies presented considerable spatiotemporal variation since the stream makes these environments more dynamic [60,61]. Alternatively, recent studies have revealed that the diversity and structure of sediment bacterial communities are not affected by space–time variations [19,24–26]. This evidence has shown that the use of a water body as a sampling matrix tends to generate a more reliable diagnosis of the diversity and structure of bacterial communities in a more restricted temporal frame, whereas sediment analyses are more suitable when the main goal is to identify chronic impacts on freshwater ecosystems.

## 5. Conclusions

Our analyses revealed that the diversity and structure of the bacterial communities of the riverbed sediment presented different patterns when we compared bacterial communities from areas under different land-use intensities. Our findings can be considered a source of evidence for the usefulness of bacterial community-based approaches as a tool for the diagnosis and monitoring of areas where human land-use is fully or partially restricted, with a purpose to guarantee the quality of fragile and vulnerable freshwater environments. These approaches can even be incorporated into regular water quality programs of river basins since the analyses of bacterial communities from riverbed sediment have been shown to be efficient for detecting chronic anthropogenic impacts on freshwater ecosystems.

**Supplementary Materials:** The following are available online at https://www.mdpi.com/article/10.3390/conservation1020008/s1: Figure S1—Rarefaction plot from the sequencing of the sediment samples by area.

**Author Contributions:** Conceptualization, G.A.L. and A.S.A.; methodology, G.A.L.; formal analysis, M.d.P.J.; investigation, M.d.P.J. and G.A.L.; resources, M.d.P.J. and T.A.C.S.; data curation, M.d.P.J. and G.A.L.; writing—original draft preparation, M.d.P.J., T.A.C.S. and G.A.L.; writing—review and editing, G.A.L. and A.S.A.; supervision, G.A.L. and A.S.A.; project administration, G.A.L.; funding acquisition, G.A.L. All authors have read and agreed to the published version of the manuscript.

**Funding:** This research was funded by Pró-Reitoria de Pesquisa, Inovação e Pós-graduação do IFMG, grant number: Edital IFMG 169/2015.

**Institutional Review Board Statement:** Not applicable.

**Informed Consent Statement:** Not applicable.

**Data Availability Statement:** All sequencing data generated in this study can be accessed from the GenBank Database at Bioproject PRJNA611749.

**Conflicts of Interest:** The authors declare no conflict of interest.

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
