# Peer review of "Assessments of Bacterial Community Shifts in Sediments along the Headwaters of São Francisco River, Brazil"

_conservation, doi:10.3390/conservation1020008_

Round 1

Reviewer 1 Report

Dear Authors,

I found the MS very interesting and well-written. However, I have some comments and suggestions to improve the MS to make it fully suitable for publication.

Key words: I would suggest to highlight the keywords more clearly, for example you have five, but some of them are almost the same. I suggest using: freshwaters, anthropogenic impact, bacterial community, water quality, bioindicators.

Introduction

In paragraph of L 44-63, please let it be clear when you use the term microbial community, microorganisms and so on, because you mix these terms throughout the text. I suggest using microbial community all the time, but if you want to refer to prokaryotes, say bacterial community. Also L46-47 this sentence is very strange because of "mainly microorganisms such as macroinvertebrates", macroinvertebrates are not only microorganisms that are within biological elements for water assessment, and what about phytoplankton and phytobenthos that are also microorganisms within biological elements. Please preformulate this sentence and make it more clear.

Materials and methods

I would suggest enlarging the letters of all the labels in the diagrams in Fig. 2.

145L Please cite the original paper in which these primers were designed.

167L Why haven't you updated the new version of Qiime 2?

Have you done all your analyses in Qiime? Also graphical representations, e.g. where did you do heatmap analyses, in which program? If you also used R for some analyses, you should mention and cite that in the MS.

For Fig. 3 the same comment as for Fig. 2, please enlarge the letters of the caption.

Fig 7 please also enlarge the letters of the taxa ranks names.

L208 and L253 You have switched the numbers of figures in the text description.

I would suggest also doing an analysis of the bacterial community in correlation with the environmental parameters, to see which parameters were significant to the community.

Discussion

L235-359 I suggest reviewing the ecology of more bacterial groups and discussing what you discovered in the samples, not just the greatest abundance. What about Actinobacteria, which is a very sensitive group of bacteria, particularly sensitive to the presence of cyanobacteria, and whose bioindicator potential has already been recognized by many researchers?

Please check your references once again, because I could not find the reference 34 in the text.

Author Response

Point 1: In paragraph of L 44-63, please let it be clear when you use the term microbial community, microorganisms and so on, because you mix these terms throughout the text. I suggest using microbial community all the time, but if you want to refer to prokaryotes, say bacterial community.

Response 1: Changes to "bacterial communities" were performed in all cases related to the term.

Point 2: Also L46-47 this sentence is very strange because of "mainly microorganisms such as macroinvertebrates", macroinvertebrates are not only microorganisms that are within biological elements for water assessment, and what about phytoplankton and phytobenthos that are also microorganisms within biological elements. Please preformulate this sentence and make it more clear.

Response 2: The suggestions were duly inserted in the text (lanes: 46-48)

Point 3: I would suggest enlarging the letters of all the labels in the diagrams in Fig. 2.

Response 3: A new Figure 2 has been done.

Point 4: 145L Please cite the original paper in which these primers were designed.

Response 4: Reference cited (lane: 164)

Point 5: 167L Why haven't you updated the new version of Qiime 2?

Response 5: We made a mistake in Qiime2 version description. The analyses were performed using Qiime2 version 2019.10. The correct description has been added to manuscript (lane: 186).

Point 6: Have you done all your analyses in Qiime? Also graphical representations, e.g. where did you do heatmap analyses, in which program? If you also used R for some analyses, you should mention and cite that in the MS.

Response 6:

The comparative analyses of water quality, presented in figure 2, were performed in R. This information was added to the figure caption.

LEfSe analysis was performed on the Galaxy web application available at https://huttenhower.sph.harvard.edu/galaxy/. This information was included in the text (lanes: 212-213).

All alpha- and beta-diversity analyses, and respective figures generated as output, were performed in Qiime2. This information was included in the text (lanes: 206-209).

Point 7: For Fig. 3 the same comment as for Fig. 2, please enlarge the letters of the caption.

Response 7: A new Figure 3 has been done.

Point 8: Fig 7 please also enlarge the letters of the taxa ranks names.

Response 8: A new Figure 7 has been done.

Point 9: L208 and L253 You have switched the numbers of figures in the text description.

Response 9: The correct sequence of the figures #4 and #5 has been adjusted in the manuscript.

Point 10: I would suggest also doing an analysis of the bacterial community in correlation with the environmental parameters, to see which parameters were significant to the community.

Response 10: Unfortunately, it was not possible to carry out the suggested environmental analyses from the sediment samples used to perform the molecular analysis (limited financial resources). Therefore, it will not be possible to satisfy this suggestion. However, we strongly believe that the absence of this suggested additional analysis will not unviable the manuscript.

Point 11: L235-359 I suggest reviewing the ecology of more bacterial groups and discussing what you discovered in the samples, not just the greatest abundance. What about Actinobacteria, which is a very sensitive group of bacteria, particularly sensitive to the presence of cyanobacteria, and whose bioindicator potential has already been recognized by many researchers?

Response 11: An additional discussion was included in the manuscript (lanes: 394-409).

Point 12: Please check your references once again, because I could not find the reference 34 in the text.

Response 12: A complete review of the references was carried out. The verified inconsistency (ref # 34) has been resolved.

Reviewer 2 Report

The manuscript entitled "Assessments of bacterial community shifts from river bed sediments constitute an useful tool for diagnosis and monitoring intensity of anthropogenic impacts " investigated the feasibility of bacterial community change as a method of river health monitoring. The idea is interesting. However, the results were not well organized and the discussion lacks evidence. Given the substantial amount of revisions needed to repair this manuscript, I suggest that the article needs a major revision before it can be published. 1-In the manuscript, the author did not effectively reveal the reason why 16s RNA is suitable as an effective method for monitoring aquatic ecological health, and did not screen characteristic microorganisms as a basis for judgment. In addition, the diversity of microorganisms in such a large watershed cannot be convinced by only 16s RNA, and environmental factors such as hydrological conditions, temperature, and rainfall need to be considered. The author needs to add relevant parts to the introduction. 2-The title needs to be revised, and the conclusion does not effectively support this title. 3-The abstract and conclusions need to be revised, and more data in the article need to be provided to confirm the conclusions reached. 4-All pictures need to be resubmitted, and the DPI of the pictures is at least 300. 5-Abbreviations need to be defined and explained, such as QIIME. In addition, the P value needs to be in italics.

Author Response

Point 1: 1- In the manuscript, the author did not effectively reveal the reason why 16s RNA is suitable as an effective method for monitoring aquatic ecological health, and did not screen characteristic microorganisms as a basis for judgment. In addition, the diversity of microorganisms in such a large watershed cannot be convinced by only 16s RNA, and environmental factors such as hydrological conditions, temperature, and rainfall need to be considered. The author needs to add relevant parts to the introduction.

Response 1: The second and third paragraphs of the Introduction section were devoted to this issue. Recent references were presented, from reliable journals, which support our argumentation. Our argument is in line with the evaluation of the other peer reviewer who indicated that the content is succinctly described and contextualized as well as questions, hypotheses and methods are clearly stated in the manuscript.

Point 2:  2-The title needs to be revised, and the conclusion does not effectively support this title

Response 2: Title and conclusion were modified.

Point 3: 3-The abstract and conclusions need to be revised, and more data in the article need to be provided to confirm the conclusions reached.

Response 3: Abstract and conclusion were modified.

Point 4: All pictures need to be resubmitted, and the DPI of the pictures is at least 300.

Response 4: All figures with the resolution indicated by the MDPI staff were submitted as separate files from the manuscript. I believe that peer reviewers did not have the opportunity to access the high resolution figures.

Point 5: 5-Abbreviations need to be defined and explained, such as QIIME. In addition, the P value needs to be in italics.

Response 5: Abbreviations were defined as recommended.

Round 2

Reviewer 2 Report

All comments have been reasonably resolved.